# A Bayesian network for modelling the Lady tasting tea experiment

Gang Xie *

Office of Research Services and Graduate Studies, Charles Sturt University, New South Wales, Australia

* gxie@csu.edu.au

## Abstract

A cup of tea can be made in one of the two ways: the milk or the tea infusion was first added to the cup. The Lady Tasting Tea experiment consists in mixing eight cups of tea, four in one way and four in the other, and presenting them to the Lady for judgment in a random order. This short article presents a Bayesian Network (BN) for modelling the Lady Tasting Tea experiment that provides a comprehensive perspective in inferential analysis of all the data samples possibly generated from the experiment. More specifically, with respect to a prior distribution of three possible levels (pure guessing, 75% sure, and 100% sure) of the Lady's ability in correctly deciding how a served cup of tea has been made, the proposed BN model enables us to calculate the posterior probabilities of any judgment outcomes possibly made by the Lady.

**Data Availability Statement:** All relevant data are within the paper.

**Funding:** The author(s) received no specific funding for this work.

## Introduction

The story and the experiment of the lady tasting tea were employed by R.A. Fisher to open the second chapter of his seminal book, 'The Design of Experiments' [1] and it is commonly regarded as the symbolic beginning of the test of significance paradigm in modern statistical analysis practice. In 2001, David Salsburg published his book 'The Lady Tasting Tea: How Statistics Revolutionized Science in the Twentieth Century,' [2] which was well-received by a wide range of readers. Twenty years later, John Richardson provided 'A closer look at the lady tasting tea' [3], in which he quoted Fisher's description of the story and the experiment [1]:

A lady declares that by tasting a cup of tea made with milk she can discriminate whether the milk or the tea infusion was first added to the cup. We will consider the problem of designing an experiment by means of which this assertion can be tested . . .

Our experiment consists in mixing eight cups of tea, four in one way and four in the other, and presenting them to the subject for judgment in a random order. The subject has been told in advance of what the test will consist, namely that she will be asked to taste eight cups, that these shall be four of each kind, and that they shall be presented to her in a random order . . . Her task is to divide the 8 cups into two sets of 4, agreeing, if possible, with the treatments received.

In order to calculate the probability or chance of the lady correctly identifying all 8 cups, assuming she had made her assessments purely by guessing (namely, the null hypothesis: Pr

**Competing interests:** The authors have declared that no competing interests exist.

(correct | each cup) = 0.5), Fisher considered the experiment consisting of 70 equally likely possible outcomes (8! / 4! / 4! = 70). Therefore, the answer is Pr(all correct | 8 cups) = 1/70 ≈ 0.0143 (namely, the *p*-value, Pr(data | hypothesis) = 0.0143 (4dp) for a one-sided test). Following this approach, it can be shown that, out of 70 all possible outcomes, there are 16 possible cases of guessing 6 cups correctly (4 x 4 = 16); 36 possible cases of guessing 4 cups correctly (6 x 6 = 36); 16 possible cases of guessing 2 cups correctly (4 x 4 = 16); and one possible case of guessing all 8 cups incorrectly. Hence, the probability distribution of the experimental outcomes can be derived and the corresponding *p*-values be calculated.

However, the very purpose of the lady tasting tea experiment is actually to assess the assumed ability level (hypothesis) of correctly identifying the cups served to this lady based on her performance (the observed sample data) and many of us would agree that this is beyond what *p*-values can do [4,5]. This motivated the development of this Bayesian network (BN) which is able to provide us the posterior probabilities for the hypothesized ability levels given any one set of the possible samples of the observed cup identification results.

## Method and results

Originally developed as a modelling tool from artificial intelligence since late 1980s, today Bayesian Networks (BNs) have found their applications range across the sciences, industries and government organizations [6,7]. Formally, a BN model is a graphical representation, i.e., a directed acyclic graph (DAG), of a joint probability distribution of a set of random variables in which each variable is represented by a node and the dependency relationship is represented by a link/edge for two associated variables [6,7]. Bayesian Network got its name because it can be considered as a mechanism for automatically applying Bayes' theorem to complex problems. The Bayes Theorem (or Bayes Rule) is a mathematical statement which expresses the interrelationships between the conditional, marginal, and joint probability distributions of random variables as defined in the following formula [8]:

$$\Pr(B|A) = \frac{\Pr(A|B)\Pr(B)}{\Pr(A)} = \frac{\Pr(A,\ B)}{\Pr(A)}, \tag{1}$$

Where A and B are two random variables/events; Pr(A) and Pr(B) are the marginal probability distributions of A and B, respectively; Pr(B|A) is the conditional probability distribution of B given A; Pr(A|B) is the conditional probability distribution of A given B; and Pr(A, B) is the joint probability distribution of A and B. In a complex model that involves many variables, through the Bayes Theorem, a BN model quantifies the local dependency relationships between a variable (node) and its parent variables (nodes) and then all local dependency relationships are integrated based on the probability chain rule so that the joint distribution of the global (i.e., overall) interrelationships among all variables can be determined/characterised [7,9]. Because a BN model represents the joint distribution of all variables included in the model, any one or a subset of these variables can be selected as the target variable(s) (equivalent to the 'response' variable in a regression model), allowing for various inferential analyses to be performed by assuming different scenarios based on the 'findings' of the remaining variables.

Essentially, a BN model follows a machine learning approach for data analysis. Although the theoretical foundation and computational algorithms underlying BNs are highly involved in subjects such as computer science, mathematics and statistics, the applications of BN models are very intuitive and relatively straightforward because of the availability of many well tested BN application software packages [7,9]. In this study, the BN model was developed using the popular BN software Netica [10] and a static picture of the model was shown in Fig 1.

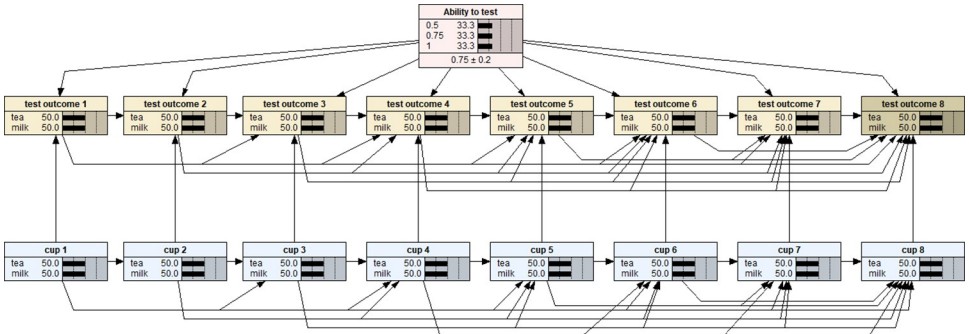

**Fig 1. A Bayesian network with an uniform prior distribution for modelling the experiment of the Lady tasting tea as described in 'The design of experiments' [1].**

Note that each variable in a BN model is represented by a node. The link between two nodes represents the dependency relationship between two variables. The middle column of each node is a percentage totalling 100%, representing the analysis outcomes of each level within a node. The last column is a graphical representation of the percentage values for each level, which are shown as distribution bars. The vertical dotted lines are markers, which are equally spaced to aid in visualising the comparative heights of the distribution bars. As shown in Fig 1, there are 17 nodes/variables in the BN model: the eight nodes in (bottom row) blue colour represent the sequence of eight cups being served each with 50% of probability being either pouring tea in first or milk first; the eight (middle row) yellow nodes represent the corresponding assessment outcome for each of the eight cups–tea first or milk first as identified by the lady; the pink node at the top is the variable specifying the prior distribution of the lady's ability to make a correct identification–the default setting is a uniform distribution with three possible ability levels: pure guessing (0.5), 100% sure (1), or an imperfect but true ability to make correct decision (0.75). Here, the concept of probability has been defined as 'propensity' while either the 'relative frequency' or 'personal belief' should only be considered as two alternative/different ways of estimating the magnitude of probability rather than the competitive metaphysical definitions of probability [11]. This BN model therefore is a joint probability distribution of all these 17 nodes/variables (with a total of 3573 conditional probability values) which fully characterises and represents the probabilistic and statistical properties of the lady tasting tea experiment. The BN model's structure has been manually specified as shown graphically in Fig 1. Note that, the 'Ability to test' and 'cup 1' are two root nodes which do not have any nodes to depend on. Once these two root nodes were set up, other nodes may be set up according to the lady tasting tea experimental design. In particular, the ways of links connecting nodes and the values input into the conditional probability tables with those eight (bottom row) blue nodes ensure that the served cups shall be four of each kind and the serve of each cup in random order; depending on the status of the prior probability node (the pink node at the top) and the (bottom row) blue nodes, the set of eight (middle row) yellow nodes enables a researcher to model the lady's tasting and identification processes. Although the default setting with the prior probability node assumes a uniform prior distribution as shown in Fig 1, the researcher can easily specify a non-uniform prior distribution for subsequent inferential analysis by employing Netica's enter finding calibration function as shown in Fig 2 [12].

With the BNs shown in Figs 1 and 2, various inferential analyses can be performed with respect to the lady tasting tea experiment. In particular, with respect to any set of served eight cups of tea (i.e., evidence results presented in (bottom row) blue nodes), the posterior probabilities for the three possible 'ability to test' options will be calculated for any set of testing

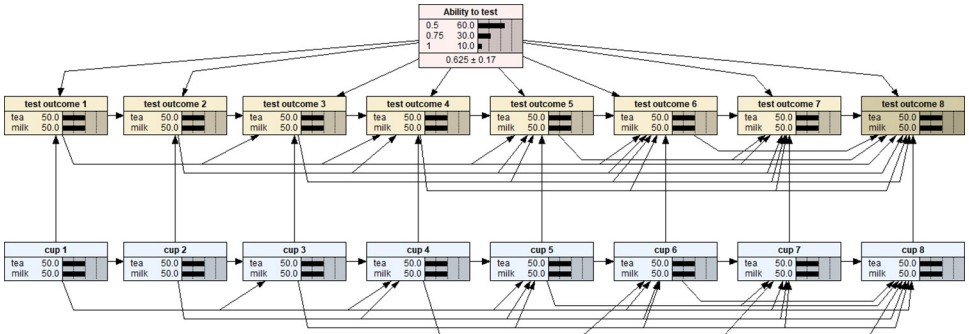

**Fig 2. The Bayesian network presented in Fig 1 being set up as a model with a non-uniform prior distribution for further inferential analyses.**

outcomes (i.e., evidence results given in (middle row) test outcome nodes). For example, suppose a testing scenario as this: the set of served eight cups of tea was TMMTMMMMTT (T = Tea first and M = Milk first in preparing the tea) and the lady correctly identified all eight cups as shown in Fig 3. The BN model with uniform prior distribution would return the posterior probability distribution as P(0.5|D) = 0.012, P(0.75|D) = 0.149, and P(1|D) = 0.839 as shown by the (top) pink node in Fig 3. That is, for this testing scenario, the observation evidence showed that the lady's ability to make a correct identification is much more likely to be 100% certain than partially certain (75:25) or totally by guessing (50:50).

Compared to a binomial trial with n = 8 experimental design, the lady tasting tea experimental design is much more difficult to model because the former consists of eight independent Bernoulli trials while the latter involves dependent trials. Therefore, different from a binomial trial design, in the lady tasting tea experiment, the posterior probability distribution depends on three factors: the order of the served cups, and both the number and the order of the correctly identified cups. For example, in Fig 4, it is a scenario that the lady supposed having correctly identified six of the eight cups served in the same order of Fig 3. The resulting posterior probability distribution were: P(0.5|D) = 0.419, P(0.75|D) = 0.581, and P(1|D) = 0. However, as shown in Fig 5, a different order of six correctly identified cups scenario ended up with a different posterior probability distribution: P(0.5|D) = 0.243, P(0.75|D) = 0.757, and P(1|D) = 0. Of course, if we would like to accept a non-uniform prior distribution BN model as in Fig 2, an extra layer of complication will add in which renders different posterior probability distributions with respect to the corresponding scenarios assumed above.

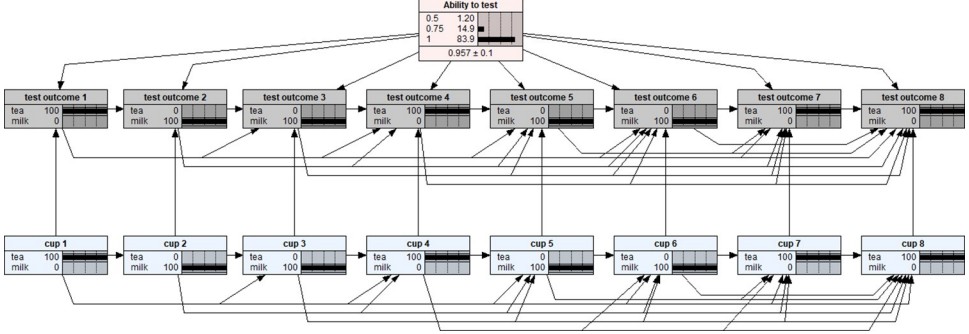

**Fig 3. The posterior probability distribution of the lady's ability to test based on a testing scenario of 100% correct identification of the served set of eight cups of tea in the order of TMMTMMTT.**

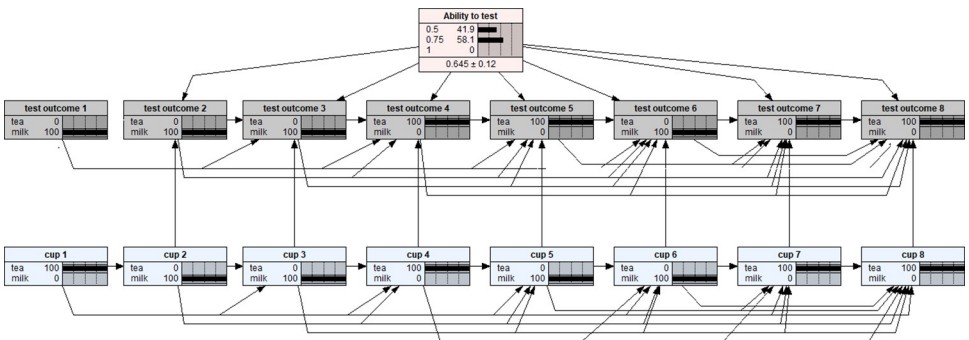

**Fig 4. The posterior probability distribution of the lady's ability to test based on a testing scenario of six of eight cups were correctly identified with the tasting identification order as MMMTTMTT (middle row test outcome nodes) against the same (as of Fig 3) served order of TMMTMMTT (bottom row blue nodes).** Hence, the tasting assessment results are: wrong, correct, correct, correct, wrong, correct, correct, correct.

## Discussion

Therefore, this article has presented a BN modelling the lady tasting tea experiment which allows us to perform various inferential analyses, in particular to obtain the posterior probability distribution for any specific set of tea tasting assessment results given a fixed pattern of serving the cups. This BN model can also play a role in providing another empirical case to show the limitations of Fisher's testing of significance and the much richer applications that the Bayesian statistical analysis can offer. To complete this short article, assuming the lady identified all eight cups correctly, a table of posterior probability distributions was presented by referring to the uniform prior distribution BN as in Fig 1 but by referring to three different settings of prior distributions. Note that, according to Fisher's test of significance paradigm, the lady in the experiment only has $1/70 \approx 0.0143 = 1.43\%$ chance to be able to get all eight cups correctly identified by pure guessing; hence, the null hypothesis (i.e., the lady making her assessment based on pure guessing only) should be rejected based on the $p$-value $< 0.05$ criterion. However, the nuances of the lady tasting tea experiment told us that the probability of achieving perfect match results could be slightly different if the set of eight cups were served in different order due to the conditional probability nature in its design; furthermore, from a Bayesian perspective, the estimation of the posterior probability of the lady's ability to make a correct decision also depended on how the prior probability distribution was specified as shown in Table 1. For example, depending on different prior distributions, the estimated posterior probability of pure guessing could be as low as 1.07% or as high as 9.69%!

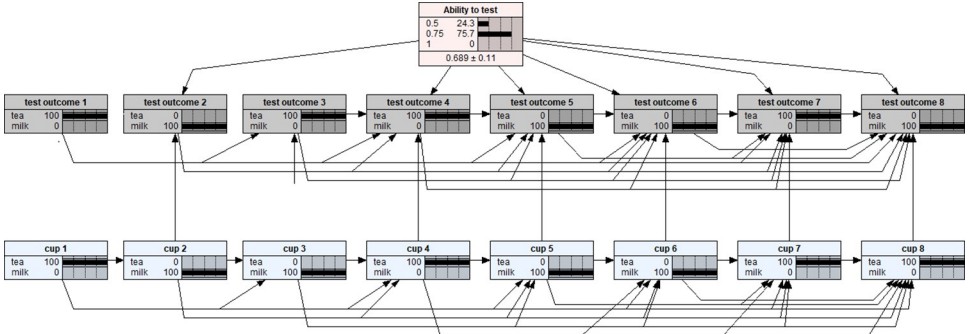

**Fig 5. The posterior probability distribution of the lady's ability to test based on a testing scenario of six of eight cups were correctly identified with the tasting identification order as TMTTMMTM (middle row test outcome nodes) against the same (as of Fig 3) served order of TMMTMMTT (bottom row blue nodes).** Hence, the tasting assessment results are: correct, correct, wrong, correct, correct, correct, correct, wrong.

**Table 1. Posterior probability distributions (in percentages) of all 70 possible sets of the making of eight cups of tea that would all be correctly identified under three prior distribution scenarios: Pr3 represents the uniform prior distribution of three possible ability levels 0.5, 0.75, 1; Pr2 represents the uniform prior distribution of binary ability levels 0.5 or 1; Pr2a represents the uniform prior distribution of binary ability levels 0.5 or 0.75 where 0.5 means a pure guessing; 0.75 means 75% of the time the making of the tea will be correctly identified and 1 means the identification will be 100% correct.**

| cup1 | cup2 | cup3 | cup4 | cup5 | cup6 | cup7 | cup8 | Pr 3 = 0.5 | Pr 3 = 0.75 | Pr 3 = 1 | Pr 2 = 0.5 | Pr 2 = 1 | Pr 2 a = 0.5 | Pr 2 a = 0.75 |
|---|---|---|---|---|---|---|---|---|---|---|---|---|---|---|
| Milk | Milk | Milk | Milk | Tea | Tea | Tea | Tea | 1.07 | 23.8 | 75.1 | 1.41 | 98.6 | 4.32 | 95.7 |
| Milk | Milk | Milk | Tea | Milk | Tea | Tea | Tea | 1.14 | 19 | 79.9 | 1.41 | 98.6 | 5.68 | 94.3 |
| Milk | Milk | Milk | Tea | Tea | Milk | Tea | Tea | 1.2 | 14.9 | 83.9 | 1.41 | 98.6 | 7.42 | 92.6 |
| Milk | Milk | Milk | Tea | Tea | Tea | Milk | Tea | 1.25 | 11.6 | 87.1 | 1.41 | 98.6 | 9.67 | 90.3 |
| Milk | Milk | Milk | Tea | Tea | Tea | Tea | Milk | 1.25 | 11.6 | 87.1 | 1.41 | 98.6 | 9.67 | 90.3 |
| Milk | Milk | Tea | Milk | Milk | Tea | Tea | Tea | 1.14 | 19 | 79.9 | 1.41 | 98.6 | 5.69 | 94.3 |
| Milk | Milk | Tea | Milk | Tea | Milk | Tea | Tea | 1.2 | 14.9 | 83.9 | 1.41 | 98.6 | 7.43 | 92.6 |
| Milk | Milk | Tea | Milk | Tea | Tea | Milk | Tea | 1.25 | 11.6 | 87.1 | 1.41 | 98.6 | 9.69 | 90.3 |
| Milk | Milk | Tea | Milk | Tea | Tea | Tea | Milk | 1.25 | 11.6 | 87.1 | 1.41 | 98.6 | 9.69 | 90.3 |
| Milk | Milk | Tea | Tea | Milk | Milk | Tea | Tea | 1.2 | 14.9 | 83.9 | 1.41 | 98.6 | 7.43 | 92.6 |
| Milk | Milk | Tea | Tea | Milk | Tea | Milk | Tea | 1.25 | 11.6 | 87.1 | 1.41 | 98.6 | 9.69 | 90.3 |
| Milk | Milk | Tea | Tea | Milk | Tea | Tea | Milk | 1.25 | 11.6 | 87.1 | 1.41 | 98.6 | 9.69 | 90.3 |
| Milk | Milk | Tea | Tea | Tea | Milk | Milk | Tea | 1.25 | 11.6 | 87.1 | 1.41 | 98.6 | 9.69 | 90.3 |
| Milk | Milk | Tea | Tea | Tea | Milk | Tea | Milk | 1.25 | 11.6 | 87.1 | 1.41 | 98.6 | 9.69 | 90.3 |
| Milk | Milk | Tea | Tea | Tea | Tea | Milk | Milk | 1.2 | 14.9 | 83.9 | 1.41 | 98.6 | 7.43 | 92.6 |
| Milk | Tea | Milk | Milk | Milk | Tea | Tea | Tea | 1.14 | 19 | 79.9 | 1.41 | 98.6 | 5.67 | 94.3 |
| Milk | Tea | Milk | Milk | Tea | Milk | Tea | Tea | 1.2 | 14.9 | 83.9 | 1.41 | 98.6 | 7.42 | 92.6 |
| Milk | Tea | Milk | Milk | Tea | Tea | Milk | Tea | 1.24 | 11.6 | 87.1 | 1.41 | 98.6 | 9.67 | 90.3 |
| Milk | Tea | Milk | Milk | Tea | Tea | Tea | Milk | 1.24 | 11.6 | 87.1 | 1.41 | 98.6 | 9.67 | 90.3 |
| Milk | Tea | Milk | Tea | Milk | Milk | Tea | Tea | 1.2 | 14.9 | 83.9 | 1.41 | 98.6 | 7.42 | 92.6 |
| Milk | Tea | Milk | Tea | Milk | Tea | Milk | Tea | 1.24 | 11.6 | 87.1 | 1.41 | 98.6 | 9.67 | 90.3 |
| Milk | Tea | Milk | Tea | Milk | Tea | Tea | Milk | 1.24 | 11.6 | 87.1 | 1.41 | 98.6 | 9.67 | 90.3 |
| Milk | Tea | Milk | Tea | Tea | Milk | Milk | Tea | 1.24 | 11.6 | 87.1 | 1.41 | 98.6 | 9.67 | 90.3 |
| Milk | Tea | Milk | Tea | Tea | Milk | Tea | Milk | 1.24 | 11.6 | 87.1 | 1.41 | 98.6 | 9.67 | 90.3 |
| Milk | Tea | Milk | Tea | Tea | Tea | Milk | Milk | 1.2 | 14.9 | 83.9 | 1.41 | 98.6 | 7.42 | 92.6 |
| Milk | Tea | Tea | Milk | Milk | Milk | Tea | Tea | 1.2 | 14.9 | 83.9 | 1.41 | 98.6 | 7.42 | 92.6 |
| Milk | Tea | Tea | Milk | Milk | Tea | Milk | Tea | 1.24 | 11.6 | 87.1 | 1.41 | 98.6 | 9.67 | 90.3 |
| Milk | Tea | Tea | Milk | Milk | Tea | Tea | Milk | 1.24 | 11.6 | 87.1 | 1.41 | 98.6 | 9.67 | 90.3 |
| Milk | Tea | Tea | Milk | Tea | Milk | Milk | Tea | 1.24 | 11.6 | 87.1 | 1.41 | 98.6 | 9.67 | 90.3 |
| Milk | Tea | Tea | Milk | Tea | Milk | Tea | Milk | 1.24 | 11.6 | 87.1 | 1.41 | 98.6 | 9.67 | 90.3 |
| Milk | Tea | Tea | Milk | Tea | Tea | Milk | Milk | 1.2 | 14.9 | 83.9 | 1.41 | 98.6 | 7.42 | 92.6 |
| Milk | Tea | Tea | Tea | Milk | Milk | Milk | Tea | 1.24 | 11.6 | 87.1 | 1.41 | 98.6 | 9.67 | 90.3 |
| Milk | Tea | Tea | Tea | Milk | Milk | Tea | Milk | 1.24 | 11.6 | 87.1 | 1.41 | 98.6 | 9.67 | 90.3 |
| Milk | Tea | Tea | Tea | Milk | Tea | Milk | Milk | 1.2 | 14.9 | 83.9 | 1.41 | 98.6 | 7.42 | 92.6 |
| Milk | Tea | Tea | Tea | Tea | Milk | Milk | Milk | 1.14 | 19 | 79.9 | 1.41 | 98.6 | 5.67 | 94.3 |
| Tea | Milk | Milk | Milk | Milk | Tea | Tea | Tea | 1.14 | 19 | 79.9 | 1.41 | 98.6 | 5.67 | 94.3 |
| Tea | Milk | Milk | Milk | Tea | Milk | Tea | Tea | 1.2 | 14.9 | 83.9 | 1.41 | 98.6 | 7.42 | 92.6 |
| Tea | Milk | Milk | Milk | Tea | Tea | Milk | Tea | 1.24 | 11.6 | 87.1 | 1.41 | 98.6 | 9.67 | 90.3 |
| Tea | Milk | Milk | Milk | Tea | Tea | Tea | Milk | 1.24 | 11.6 | 87.1 | 1.41 | 98.6 | 9.67 | 90.3 |
| Tea | Milk | Milk | Tea | Milk | Milk | Tea | Tea | 1.2 | 14.9 | 83.9 | 1.41 | 98.6 | 7.42 | 92.6 |
| Tea | Milk | Milk | Tea | Milk | Tea | Milk | Tea | 1.24 | 11.6 | 87.1 | 1.41 | 98.6 | 9.67 | 90.3 |
| Tea | Milk | Milk | Tea | Milk | Tea | Tea | Milk | 1.24 | 11.6 | 87.1 | 1.41 | 98.6 | 9.67 | 90.3 |
| Tea | Milk | Milk | Tea | Tea | Milk | Milk | Tea | 1.24 | 11.6 | 87.1 | 1.41 | 98.6 | 9.67 | 90.3 |
| Tea | Milk | Milk | Tea | Tea | Milk | Tea | Milk | 1.24 | 11.6 | 87.1 | 1.41 | 98.6 | 9.67 | 90.3 |

*(Continued)*

**Table 1.** (Continued)

| cup1 | cup2 | cup3 | cup4 | cup5 | cup6 | cup7 | cup8 | Pr 3 = 0.5 | Pr 3 = 0.75 | Pr 3 = 1 | Pr 2 = 0.5 | Pr 2 = 1 | Pr 2 a = 0.5 | Pr 2 a = 0.75 |
|---|---|---|---|---|---|---|---|---|---|---|---|---|---|---|
| Tea | Milk | Milk | Tea | Tea | Tea | Milk | Milk | 1.2 | 14.9 | 83.9 | 1.41 | 98.6 | 7.42 | 92.6 |
| Tea | Milk | Tea | Milk | Milk | Milk | Tea | Tea | 1.2 | 14.9 | 83.9 | 1.41 | 98.6 | 7.42 | 92.6 |
| Tea | Milk | Tea | Milk | Milk | Tea | Milk | Tea | 1.24 | 11.6 | 87.1 | 1.41 | 98.6 | 9.67 | 90.3 |
| Tea | Milk | Tea | Milk | Milk | Tea | Tea | Milk | 1.24 | 11.6 | 87.1 | 1.41 | 98.6 | 9.67 | 90.3 |
| Tea | Milk | Tea | Milk | Tea | Milk | Milk | Tea | 1.24 | 11.6 | 87.1 | 1.41 | 98.6 | 9.67 | 90.3 |
| Tea | Milk | Tea | Milk | Tea | Milk | Tea | Milk | 1.24 | 11.6 | 87.1 | 1.41 | 98.6 | 9.67 | 90.3 |
| Tea | Milk | Tea | Milk | Tea | Tea | Milk | Milk | 1.2 | 14.9 | 83.9 | 1.41 | 98.6 | 7.42 | 92.6 |
| Tea | Milk | Tea | Tea | Milk | Milk | Milk | Tea | 1.24 | 11.6 | 87.1 | 1.41 | 98.6 | 9.67 | 90.3 |
| Tea | Milk | Tea | Tea | Milk | Milk | Tea | Milk | 1.24 | 11.6 | 87.1 | 1.41 | 98.6 | 9.67 | 90.3 |
| Tea | Milk | Tea | Tea | Milk | Tea | Milk | Milk | 1.2 | 14.9 | 83.9 | 1.41 | 98.6 | 7.42 | 92.6 |
| Tea | Milk | Tea | Tea | Tea | Milk | Milk | Milk | 1.14 | 19 | 79.9 | 1.41 | 98.6 | 5.67 | 94.3 |
| Tea | Tea | Milk | Milk | Milk | Milk | Tea | Tea | 1.2 | 14.9 | 83.9 | 1.41 | 98.6 | 7.43 | 92.6 |
| Tea | Tea | Milk | Milk | Milk | Tea | Milk | Tea | 1.25 | 11.6 | 87.1 | 1.41 | 98.6 | 9.69 | 90.3 |
| Tea | Tea | Milk | Milk | Milk | Tea | Tea | Milk | 1.25 | 11.6 | 87.1 | 1.41 | 98.6 | 9.69 | 90.3 |
| Tea | Tea | Milk | Milk | Tea | Milk | Milk | Tea | 1.25 | 11.6 | 87.1 | 1.41 | 98.6 | 9.69 | 90.3 |
| Tea | Tea | Milk | Milk | Tea | Milk | Tea | Milk | 1.25 | 11.6 | 87.1 | 1.41 | 98.6 | 9.69 | 90.3 |
| Tea | Tea | Milk | Milk | Tea | Tea | Milk | Milk | 1.2 | 14.9 | 83.9 | 1.41 | 98.6 | 7.43 | 92.6 |
| Tea | Tea | Milk | Tea | Milk | Milk | Milk | Tea | 1.25 | 11.6 | 87.1 | 1.41 | 98.6 | 9.69 | 90.3 |
| Tea | Tea | Milk | Tea | Milk | Milk | Tea | Milk | 1.25 | 11.6 | 87.1 | 1.41 | 98.6 | 9.69 | 90.3 |
| Tea | Tea | Milk | Tea | Milk | Tea | Milk | Milk | 1.2 | 14.9 | 83.9 | 1.41 | 98.6 | 7.43 | 92.6 |
| Tea | Tea | Milk | Tea | Tea | Milk | Milk | Milk | 1.14 | 19 | 79.9 | 1.41 | 98.6 | 5.69 | 94.3 |
| Tea | Tea | Tea | Milk | Milk | Milk | Milk | Tea | 1.25 | 11.6 | 87.1 | 1.41 | 98.6 | 9.67 | 90.3 |
| Tea | Tea | Tea | Milk | Milk | Milk | Tea | Milk | 1.25 | 11.6 | 87.1 | 1.41 | 98.6 | 9.67 | 90.3 |
| Tea | Tea | Tea | Milk | Milk | Tea | Milk | Milk | 1.2 | 14.9 | 83.9 | 1.41 | 98.6 | 7.42 | 92.6 |
| Tea | Tea | Tea | Milk | Tea | Milk | Milk | Milk | 1.14 | 19 | 79.9 | 1.41 | 98.6 | 5.68 | 94.3 |
| Tea | Tea | Tea | Tea | Milk | Milk | Milk | Milk | 1.07 | 23.8 | 75.1 | 1.41 | 98.6 | 4.32 | 95.7 |

It is hoped that readers could taste the richness of the inferential analysis results possibly obtained from the lady tasting tea experiment. Furthermore, interested readers may employ the BN model attached to this article to explore more inferential analyses by assuming different scenarios (of whatever their research interest) that are permitted by the lady tasting tea experiment. While the posterior probabilities of the Lady Tasting Tea experiment can be obtained from the BN model proposed in this study, expanding the current model to include more flexible prior distributions presents a challenging task. For example, it would be more realistic to assume that the Lady has more than three ability levels, leading to more realistic posterior probability patterns. This should be considered a limitation of this study and warrants future research work.

## Supporting information

**S1 File. The Bayesian network model presented in this article is provided as a supporting information file.** The model is saved as "The Lady Tasting Tea_extension.dne".
(DNE)

## Author Contributions

**Conceptualization:** Gang Xie.

**Formal analysis:** Gang Xie.

**Investigation:** Gang Xie.

**Methodology:** Gang Xie.

**Writing – original draft:** Gang Xie.

**Writing – review & editing:** Gang Xie.

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
