## [Decision Letter · Decision Letter 0]

4 Jun 2024

PONE-D-24-03996A Bayesian Network for modelling the Lady Tasting Tea experimentPLOS ONE

Dear Dr. Xie,

Thank you for submitting your manuscript to PLOS ONE. After careful consideration, we feel that it has merit but does not fully meet PLOS ONE’s publication criteria as it currently stands. Therefore, we invite you to submit a revised version of the manuscript that addresses the points raised during the review process.

We look forward to receiving your revised manuscript.

Kind regards,

Gonzalo A. Ruz, Ph.D.

Academic Editor

PLOS ONE

Journal Requirements:

Additional Editor Comments:

It is important that the author be able to comply with what was requested by the reviewers, in particular, what was indicated by reviewer 2, in relation to being able to provide more details on the methodology and experimentation for reproducibility purposes.

Reviewers' comments:

Reviewer's Responses to Questions

**Comments to the Author**

1. Is the manuscript technically sound, and do the data support the conclusions?

Reviewer #1: Yes

Reviewer #2: Partly

2. Has the statistical analysis been performed appropriately and rigorously? 

Reviewer #1: Yes

Reviewer #2: I Don't Know

3. Have the authors made all data underlying the findings in their manuscript fully available?

Reviewer #1: Yes

Reviewer #2: Yes

4. Is the manuscript presented in an intelligible fashion and written in standard English?

Reviewer #1: Yes

Reviewer #2: Yes

5. Review Comments to the Author

Reviewer #1: PLOS ONE

A Bayesian Network for modelling the Lady Tasting Tea experiment

The authors present research on the use of BN modelling with Netica BN software using the lady tasting tea experiment—a sensory discrimination test (‘M+N’ method with M=N=4, i.e., the octad method or double-tetrad) as an advantage over Fisher’s exact one-sided test assuming equal probability (p-value=0.0143) for each trial. Motivation for this approach was to provide posterior probabilities for hypothesized ability level given any set of possible samples of the observed results—thus assessing the assumed ability level of correct identification based on observed performance vs basis on p-values.

BN models (joint probability distribution of 17 nodes) are depicted in Figures 1-5 for illustration where the top node (1) represents the assumed prior distribution ‘ability to test’ (either uniform or non-uniform), the bottom nodes (8) represent the sequence of samples (tea or milk first), and the middle nodes (8) represent the corresponding assessment outcomes--applying ‘propensity’ as the concept of probability; the posterior probability distribution and ability to test are depicted at the top node--based on serving order and sample identification results. Inferential statistics can then be applied for any set of served cups with posterior probabilities calculated for any set of outcomes. The dependent (vs independent) nature of this approach is noted (i.e., n=8 independent Bernoulli trials vs ‘octad’ experimental design)—3 dependent factors were identified and exemplified via examples and discussion: order of served cups, both number and order of correctly identified cups, plus prior distribution assumption (uniform or non-uniform).

I believe the authors have done an excellent job in describing an empirical case of the lady tasting tea experiment and exemplifying their use of the BN model in this paper. Figures 1-5 and the Table 1 are clear and well presented. The paper is well organized and easy to read/follow. I have no challenges or questions in regard to the application as highlighted in this paper. The authors may suggest follow-on research to expand upon their ‘hopeful’ acceptance/adoption of the method as another case of limitations of Fisher’s testing of significance. Also, they should call out limitations for this BN modelling approach.

Suggested Edits

INTRODUCTION

Needs reference:

In 2001, David Salsburg published his book ‘The Lady Tasting Tea: How

Statistics Revolutionized Science in the Twentieth Century,’ which was well-received by a

wide range of reader

Lines 81-82 –

Drop can--“can fully characterises and represents the probabilistic and statistical properties of the lady tasting tea experiment.”

METHOD AND RESULTS

Line 85 – change “maybe” to “may be”

DISCUSSION

Line 153 – “Therefore” not needed

Lines 153-154 -- Drop “that”:

Lines 156-158 “Hopefully this BN model can…” – Perhaps state this more positively vs hedging—“This BN model can…”

“Therefore, this article has presented a BN that modelling the lady tasting tea experiment which allows us to perform…”

to

“Therefore, this article has presented a BN modelling the lady tasting tea experiment which allows us to perform…”

Line 162 – comma after paradigm

Line 167 – Change to “were served”

Line 171 – Change to “a different prior distribution”, OR “depending on different prior distributions”

Lines 206-207—Bayesian network model in supporting information—I was not able to access this BN model that is supposed to be in the supporting information to check/assess it.

Reviewer #2: The article seems to have been written somewhat carelessly. It would be good to organize the article and review the notations used (in detail). It is not clear how reproducible and reusable the experiment is. It would be helpful to provide more details about how the program works.

6. PLOS authors have the option to publish the peer review history of their article (what does this mean?). If published, this will include your full peer review and any attached files.

Reviewer #1: No

Reviewer #2: No

---

## [Author Response · Author response to Decision Letter 0]

10 Jun 2024

7 June, 2024

To: Gonzalo A. Ruz, Ph.D.

 Academic Editor

 PLOS ONE

Dear Dr. Gonzalo A. Ruz,

Thank you for the opportunity of revision of our manuscript. We really appreciate the time and effort you and the reviewers committed in reviewing our manuscript and providing the valuable feedback for the revision. We have carefully considered all the comments and have made the necessary revisions to improve the quality of our work. Below, we provide a detailed, point-by-point response to each comment. The reviewers' comments are included in italics, followed by our responses in regular font. 

Reviewer #1:

Comment 1: “I believe the authors have done an excellent job in describing an empirical case of the lady tasting tea experiment and exemplifying their use of the BN model in this paper. Figures 1-5 and the Table 1 are clear and well presented. The paper is well organized and easy to read/follow. I have no challenges or questions in regard to the application as highlighted in this paper.”

Response: We are sincerely appreciate your positive feedback and delighted to hear that you found our manuscript to be well-organized, clear, and easy to follow. We are pleased that our description of the empirical case and the use of the BN model met your expectations. Your recognition of the clarity of Figures 1-5 and Table 1 is very encouraging. Thank you for your kind words, which motivate us to continue our research with the same rigor and clarity.

Comment 2: “The authors may suggest follow-on research to expand upon their ‘hopeful’ acceptance/adoption of the method as another case of limitations of Fisher’s testing of significance. Also, they should call out limitations for this BN modelling approach.”

Response: Change has been made by adding a few sentences at the end of Discussion section to suggest directions for follow-on research.

Comment 3: “Suggested Edits

INTRODUCTION

Needs reference:

In 2001, David Salsburg published his book ‘The Lady Tasting Tea: How

Statistics Revolutionized Science in the Twentieth Century,’ which was well-received by a

wide range of reader

Lines 81-82 –

Drop can--“can fully characterises and represents the probabilistic and statistical properties of the lady tasting tea experiment.”

METHOD AND RESULTS

Line 85 – change “maybe” to “may be”

DISCUSSION

Line 153 – “Therefore” not needed

Lines 153-154 -- Drop “that”:

Lines 156-158 “Hopefully this BN model can…” – Perhaps state this more positively vs hedging—“This BN model can…”

“Therefore, this article has presented a BN that modelling the lady tasting tea experiment which allows us to perform…”

to

“Therefore, this article has presented a BN modelling the lady tasting tea experiment which allows us to perform…”

Line 162 – comma after paradigm

Line 167 – Change to “were served”

Line 171 – Change to “a different prior distribution”, OR “depending on different prior distributions” ”

Response: Thank you for your thorough editing check of our manuscript. We have carefully examined all your suggestions and have incorporated them into the revised version.

Comment 4: “Lines 206-207—Bayesian network model in supporting information—I was not able to access this BN model that is supposed to be in the supporting information to check/assess it.”

Response: We have indeed submitted the BN model file as part of the supporting information when the manuscript was first submitted. You may check with the Editor to request the submitted BN model for assessment. Otherwise, we are more than happy to provide the BN to you directly upon receiving permission of the journal Editor. 

Reviewer #2:

Comment 1: “The article seems to have been written somewhat carelessly. It would be good to organize the article and review the notations used (in detail). It is not clear how reproducible and reusable the experiment is. It would be helpful to provide more details about how the program works.”

Response: We thank this reviewer for his/her time and effort in reviewing our manuscript and providing the valuable feedback. Since the feedback was generally broad, we have done our best to respond in a more specific manner and hope our responses meet reviewer’s expectations. 

The structure and style of our manuscript may not follow exactly the common pattern of a research project journal paper because the very purpose of this study is to provide a novel Bayesian solution to the classic Frequentist hypothesis testing question based on the well-known the Lady Tasting Tea experiment that was first introduced by R. A. Fisher some 100 years ago. In the writing of the manuscript, we implicitly assumed that the target readers were familiar with major statistical paradigms (e.g., frequentist and Bayesian approaches) and Bayesian network (BN). This assumption may have been unrealistic. To address the reviewer’s concern, we have therefore added a few paragraphs in the Method and Results section to give a brief introduction to the theoretical foundation of Bayesian network and how BN works for statistical analysis. 

We have employed standard mathematical and statistical notations consistent with year-one undergraduate-level textbooks. However, the concepts such as prior and posterior probabilities may be new to readers who are not familiar with Bayesian statistics. The distinction between different probability definitions (e.g., relative frequency definition versus propensity definition) can be both philosophical and challenging to appreciate. 

The analyses conducted using the BN model and the results presented in the manuscript can be fully reproduced. We have indeed submitted the BN model file as part of the supporting information when the manuscript was first submitted. You may check with the Editor to request the submitted BN model for assessment. Otherwise, we are more than happy to provide the BN to you directly upon receiving permission of the journal Editor. 

All the changes made to the manuscript can be identified clearly in the marked-up copy of the revised version of our manuscript.

Thank you again for your time and effort in handling our manuscript.

Sincerely,

Gang Xie

Charles Sturt University, Australia

---

## [Decision Letter · Decision Letter 1]

15 Jul 2024

A Bayesian Network for modelling the Lady Tasting Tea experiment

PONE-D-24-03996R1

Dear Dr. Xie,

We’re pleased to inform you that your manuscript has been judged scientifically suitable for publication and will be formally accepted for publication once it meets all outstanding technical requirements.

Kind regards,

Gonzalo A. Ruz, Ph.D.

Academic Editor

PLOS ONE

Additional Editor Comments (optional):

Both reviewers are satisfied with the revised version of the manuscript.

Reviewers' comments:

Reviewer's Responses to Questions

**Comments to the Author**

1. If the authors have adequately addressed your comments raised in a previous round of review and you feel that this manuscript is now acceptable for publication, you may indicate that here to bypass the “Comments to the Author” section, enter your conflict of interest statement in the “Confidential to Editor” section, and submit your "Accept" recommendation.

Reviewer #1: All comments have been addressed

Reviewer #2: All comments have been addressed

2. Is the manuscript technically sound, and do the data support the conclusions?

Reviewer #1: Yes

Reviewer #2: Yes

3. Has the statistical analysis been performed appropriately and rigorously? 

Reviewer #1: Yes

Reviewer #2: Yes

4. Have the authors made all data underlying the findings in their manuscript fully available?

Reviewer #1: Yes

Reviewer #2: Yes

5. Is the manuscript presented in an intelligible fashion and written in standard English?

Reviewer #1: Yes

Reviewer #2: Yes

6. Review Comments to the Author

Reviewer #1: I am satisfied with the revision of this paper, though I did not pursue the supplementary file from the editor as I don't have access to the Netica software used (I did not download the free LIMITED version). I trust the supplementary file is complete in order for someone to replicate the research herein. The additional explanation of the methodology should help, though I believe some background in Bayesian analysis makes the paper more accessible.

Reviewer #2: All observations have been addressed, therefore I recommend the manuscript "A Bayesian Network for modelling the Lady Tasting Tea experiment" for publication in PLOS ONE.

7. PLOS authors have the option to publish the peer review history of their article (what does this mean?). If published, this will include your full peer review and any attached files.

Reviewer #1: No

Reviewer #2: No

---

## [Editor Report · Acceptance letter]

17 Jul 2024

PONE-D-24-03996R1 

PLOS ONE

Dear Dr. Xie, 

I'm pleased to inform you that your manuscript has been deemed suitable for publication in PLOS ONE. Congratulations! Your manuscript is now being handed over to our production team.

Kind regards, 

on behalf of

Prof. Gonzalo A. Ruz 

Academic Editor

PLOS ONE